# High-Speed Temperature Control Method for MEMS Thermal Gravimetric Analyzer Based on Dual Fuzzy PID Control

**DOI:** 10.3390/mi14050929

**Published:** 2023-04-25

**Authors:** Xiaoyang Zhang, Zhi Cao, Shanlai Wang, Lei Yao, Haitao Yu

**Affiliations:** 1School of Microelectronics, Shanghai University, Shanghai 200444, China; kanolin@shu.edu.cn (X.Z.); wangshanlai@shu.edu.cn (S.W.); 2State Key Laboratory of Transducer Technology, Shanghai Institute of Microsystem and Information Technology, Chinese Academy of Sciences, Shanghai 200050, China; 216062101@mail.sit.edu.cn; 3School of Chemical and Environmental Engineering, Shanghai Institute of Technology, Shanghai 201418, China

**Keywords:** temperature control, thermal gravimetric analyzer, resonant cantilever, fuzzy control, PID control, MEMS TGA

## Abstract

The traditional thermal gravimetric analyzer (TGA) has a noticeable thermal lag effect, which restricts the heating rate, while the micro-electro-mechanical system thermal gravimetric analyzer (MEMS TGA) utilizes a resonant cantilever beam structure with high mass sensitivity, on-chip heating, and a small heating area, resulting in no thermal lag effect and a fast heating rate. To achieve high-speed temperature control for MEMS TGA, this study proposes a dual fuzzy proportional-integral-derivative (PID) control method. The fuzzy control adjusts the PID parameters in real-time to minimize overshoot while effectively addressing system nonlinearities. Simulation and actual testing results indicate that this temperature control method has a faster response speed and less overshoot compared to traditional PID control, significantly improving the heating performance of MEMS TGA.

## 1. Introduction

The thermal gravimetric analyzer (TGA) is widely utilized in research and development, process optimization, and quality control in various fields such as minerals, inorganic materials, organic materials, ceramics, metal oxides, polymers, electronic materials, drugs, and food. They can effectively guide production in the analysis and control of industrial processes [1,2,3]. 

Traditional TGAs mainly consist of a high-precision electronic balance, a heating furnace, and a thermocouple. The MEMS TGA uses an integrated silicon resonant micro-cantilever chip to implement all the functions of a traditional TGA [4]. During the heating process of a traditional TGA, with the heating furnace, the internal temperature is not uniform but shows a certain temperature gradient [5]. The faster the heating rate, the greater the temperature gradient, the more difficult it is for the sample to be evenly heated, and the higher the decomposition temperature of the sample, resulting in the thermal lag phenomenon of the traditional TGA [6]. The heating furnace requires a large internal heating space and a large temperature change range, resulting in temperature controls with lag, nonlinearity, and uncertainty, making it difficult to model the system [7]. The heating method of the MEMS TGA differs from the traditional furnace heating method. It integrates a micro-heater on a resonant micro-cantilever and adopts an on-chip heating method, which enables temperature control with the advantages of fast response, low power consumption, and no hysteresis. At the same time, the consumed sample mass is about 5–50 mg for traditional TGAs [8,9]. The more samples are used, the more difficult it is to uniformly heat them, and only the heating rate can be sacrificed to ensure accuracy [10,11]. MEMS TGA directly characterizes mass changes through the resonance frequency, which guarantees high precision with a small sample size [12,13]. In summary, traditional TGAs usually use a high-temperature furnace tube to heat the sample, and the heating and cooling rates are relatively slow, which cannot meet the research needs of rapid thermal processing and thermal shock, and has reached a technical bottleneck [14]. Due to its ability to rapidly heat and cool, the MEMS TGA has significant advantages in the research of thermal cracking [15,16,17], annealing [18,19,20], and thermal shock [21,22,23], which can achieve research goals that a traditional TGA cannot achieve. The MEMS TGA can be widely used in materials science, chemistry, energy, and other fields, providing an important means to explore the thermal and dynamic characteristics of materials.

Currently, the temperature control of the MEMS sensors is mainly to suppress the thermal drift of the device output signal, keeping the temperature control at a fixed value [24,25,26]. However, the temperature control of the MEMS TGA is to achieve the program heating function. There is a significant difference between the objects being controlled by a traditional TAG and a MEMS TGA. The MEMS TGA adopts a micro-cantilever chip as the core component, enabling rapid temperature control during thermal analysis. However, traditional PID temperature control methods may lead to issues such as insufficient heating rates and temperature overshoots, which need to be addressed. The fuzzy PID control algorithm provides an effective technical solution to tackle this problem. For example, Jiang et al. [27] designed a MEMS-integrated heater temperature control system using a fuzzy PID control algorithm to control the temperature, which can be used with various MEMS devices that are easily affected by thermal drift, such as accelerometers, gyroscopes, and pressure gauges, to keep the sensors at a constant temperature. This method utilizes a thermally conductive adhesive to attach the heater and the control sensor together, which can serve as a reference but is not applicable to cantilever beam structures. Chen et al. [28] proposed a method to obtain PID initial parameters using genetic algorithms and improved the fuzzy PID control parameters. These methods all adopt the PID control and on-chip heating methods. However, constant temperature control is different from program control temperature, the temperature change span of program control temperature is large. Lu et al. [29] used a long-short-term memory network to model the system and selected the optimized PID parameters through particle swarm algorithms. These two methods have greatly improved the temperature control and are superior to the traditional PID control method. However, for the program temperature control required by the thermogravimetric analyzer, the temperature variation span is large, and these methods cannot effectively deal with the system’s nonlinearity. Chen et al. [30] introduced output quantity into the fuzzy inference and added nonlinear correction parameters, effectively accelerating the temperature control speed of the TGA in the high-temperature stage, which greatly improved the nonlinear temperature problem.

In this paper, the small size of the integrated heater and its unique thermal insulation structure design allows the cantilever beam’s free end to achieve a theoretical maximum heating rate of 10^5^ °C/s. To accurately control this heater, a dual fuzzy PID control strategy was used to design a high-speed temperature control system, effectively solving the problems of the system’s nonlinearity and large overshoot. Through simulation and experimental measurements, the significant advantages of the dual fuzzy PID control, in high-speed temperature control relative to traditional PID control, are verified. Table 1 below shows a comparison of this method with other literature in terms of temperature control rate.

## 2. Principle of Integrated Microheater

The integrated microheater is located at the free end of the cantilever’s beam and reserves a section of space at the free end for sample coating. In the cantilever with an integrated heater, a through-hole is made at the rear end, dividing the cantilever into high and low-temperature areas, preventing heat conduction and dissipation. The length, width, and thickness of the cantilever are set to 290 μm, 140 μm, and 3 μm, respectively. The hole structure is designed as a square with a side length of 100 μm. The thickness of the molybdenum resistor in the integrated heater is 100 angstroms (100 Å, 10 nm), with a width of 3.5 μm and a spacing of 3.5 μm.

The microheater leads out four paths, as shown in Figure 1, which is beneficial for the construction of the peripheral control circuit. The peripheral circuit employs a highly precise method for measuring resistance, known as the four-wire resistance measurement [31]. The material of the heater is a high-temperature resistant metal, molybdenum. The resistance of the molybdenum changes linearly with the temperature, which facilitates subsequent temperature control and processing. The formula for the resistance change rate is as follows:(1)$$R(T)=R_{0}\times (1+\alpha \times (T-T_{0}))$$
where *T* is the temperature value in Celsius, *R*(*T*) denotes the resistance value at temperature *T*, *R*_0_ signifies the resistance value at temperature *T*_0_, and the temperature coefficient *α* (in units of 1/°C) is a constant. The temperature measurements were conducted using an infrared thermometer, and corresponding temperature and resistance values were obtained during the heating process. Through linear fitting, it was determined that the resistance *R*_0_ at *T*_0_ = 0 °C is 295.1 Ω, and the temperature coefficient *α* is 0.9158 Ω/°C.

The thermal simulation results are shown in Figure 2, which demonstrates a uniform temperature distribution in the sample area.

## 3. Design of Temperature Control Method

Through studying the temperature characteristics of the micro-integrated heaters in this section, the data from the open-loop step response experiment were imported into the MATLAB system identification toolbox to identify the control system’s transfer function. The PID control and two fuzzy controllers were used to adjust the PID control parameters in real-time.

### 3.1. Transfer Function of Controlled Object

To simulate the controlled object, using Simulink, it is necessary to determine the transfer function of the object. This article analyzes the entire process of temperature control for an integrated heater, primarily composed of the following variables: thermistor surface area *A*, thermistor mass *m*, external air temperature *T_ambient_*, heat generated by the thermistor *Q_in_*, heat dissipated *Q_out_*, heat transfer coefficient *h*, the specific heat capacity of the thermistor *c_p_*, and real-time thermistor temperature *T*(*t*).

The heat balance equation is:(2)$$m\times c_{p}\times \frac{dT(t)}{dt}=Q_{in}-Q_{out}$$
where
(3)$$Q_{out}=h\times A\times (T(t)-T_{\mathrm{ambient}})$$

Substituting *Q_out_* into the heat balance equation, we get:(4)$$m\times c_{p}\times \frac{dT(t)}{dt}=Q_{in}-h\times A\times (T(t)-T_{\mathrm{ambient}})$$

To obtain the transfer function, we need to convert this differential equation into the Laplace domain. Let *s* be the Laplace variable, *T*(*s*) be the Laplace transform of *T*(*t*), and *Q*_*in*_(*s*) be the Laplace transform of *Q_in_*. By the properties of the Laplace transform, we have:(5)$$m\times c_{p}\times s\times T(s)=Q_{in}(s)-h\times A\times (T(s)-T_{\mathrm{ambient}})$$

Rearranging this equation, we can obtain the transfer function:(6)$$G(s)=\frac{T(s)}{Q_{in}(s)}=\frac{1}{m\times c_{p}\times s+h\times A}$$

This transfer function represents the transient response of the thermistor system. Note that this is a first-order transfer function with a single real pole, meaning that the system’s transient response will decay exponentially to a steady-state value.

To account for the effect of the thermistor temperature changes causing resistance changes, it is necessary to introduce Equation (1).

The power on the resistor is proportional to the square of the voltage, i.e.,
(7)$$P=\frac{V^{2}}{R(T)}$$
where *P* is the power on the thermistor, and *V* is the step voltage. Substituting *R*(*T*) with *P*, we get:(8)$$P=\frac{V^{2}}{R_{0}\times (1+\alpha \times (T-T_{0}))}$$

Now, we need to modify the heat balance equation to:(9)$$m\times c_{p}\times \frac{dT(t)}{dt}=P-h\times A\times (T(t)-T_{\mathrm{ambient}})$$

Substituting the expression for *P*, we get:(10)$$m\times c_{p}\times \frac{dT(t)}{dt}=\frac{V^{2}}{R_{0}\times (1+\alpha \times (T(t)-T_{0}))}-h\times A\times (T(t)-T_{\mathrm{ambient}})$$

To obtain the transfer function, we need to convert this nonlinear differential equation into the Laplace domain. Let vs. be the Laplace transform of the step voltage, and *T*(*s*) be the Laplace transform of *T*(*t*). Solving in the Laplace domain becomes complex due to the nonlinearity of the equation. In this case, alternative methods, such as nonlinear system identification techniques, might be required to obtain the transfer function.

The Hammerstein model is a nonlinear system identification method, where a nonlinear static system (usually represented by a nonlinear function) is cascaded with a linear dynamic system (represented by a linear transfer function). This model simplifies the representation of a nonlinear system, making it easier to analyze and control. The mathematical representation of the Hammerstein model is as follows:(11)y(t)=G[f(u(t))]=∫[h(τ)∗f(u(t−τ))]dτ
where *y*(*t*) is the system’s output, *u*(*t*) is the system’s input, *f* is the nonlinear function (usually represented by a polynomial or neural network), *h*(*τ*) is the linear system’s impulse response, *G* is the linear transfer function, and * denotes convolution.

To implement the Hammerstein model in Simulink, we use two subsystems, one representing the nonlinear static system and the other representing the linear dynamic system. First, we connect the input signal to the nonlinear static system, and then connect the output of the nonlinear static system to the input of the linear dynamic system. The output of the linear dynamic system will be the system’s output.

In this paper, the integrated heater is both a temperature control resistor and a temperature measurement resistor, and the transfer function of the controlled object is complex. It is difficult to determine the form of the transfer function through theoretical deduction, so we use the System Identification Toolbox in MATLAB to automatically identify the transfer function of the controlled object through open-loop step response experiments. The input voltage measured in the standby state of the temperature control system is 748.8 mV, with a resistance of 483.8 Ω and a sampling time of 0.1 s. By replacing the input voltage with 4697.8 mV, the system reaches a steady state after about 160 s, and the response curve is shown in Figure 3. The identified transfer function of the controlled object is shown in the following equation:(12)$$G(s)=\frac{0.1342s^{2}+0.04127s+0.0008969}{s^{2}+0.2798s+0.005739}$$

The transfer function obtained is a second-order model with a fitting accuracy of 93.13%. The goodness of fit is commonly represented by the R-squared value *R*^2^ or the coefficient of determination, which ranges from zero to one. The *R*^2^ value indicates the degree to which the model fits the data, i.e., the difference between the fitted curve and the actual data. The closer the *R*^2^ value is to one, the higher the degree of correspondence between the fitted results and the actual data, and the stronger the model’s explanatory power for the data. Conversely, when the *R*^2^ value is close to zero, it indicates that the model has a weaker explanatory power for the data. The appropriateness of the transfer function’s goodness of fit depends on the specific application and research purpose. In practice, a high goodness of fit (e.g., *R*^2^ value greater than 0.9 or 0.95) is generally considered to be a good fitting result. Based on actual experimental results, this level of goodness of fit is sufficient to simulate the controlled object. The calculation method for the R-squared value is as follows: (13)$$R^{2}=1-\frac{SSR}{SST}$$
where *SSR* represents the sum of squared residuals (the sum of squared differences between the fitted curve and the actual data), and *SST* represents the total sum of the squares (the sum of squared differences between the actual data and the mean).

### 3.2. Select a Temperature Control Scheme

The PID control structure is simple and easy to implement, widely used in industrial process controls. The algorithm combines the current output value and the deviation from the target value by combining proportional, integral, and differential components to produce an output applied to the controlled object. The error calculation formula is as follows:(14)E(t)=R(t)−Y(t)
where *E*(*t*) represents the deviation between the target value and the output value at time *t*, *R*(*t*) represents the target value at time *t*, and *Y*(*t*) represents the output value of the controlled object at time *t*.

The continuous time signals are sampled and quantized, converting the integral into a summation form and the differential into a different form, resulting in the discrete form of the PID control algorithm as follows:(15)$$U\left(t\right)=K_{p}\times [E\left(t\right)+\frac{T_{s}}{T_{i}}\times \sum_{k=0}^{\infty }\;E_{K}+\frac{T_{d}}{T_{s}}\times EC\left(t\right)]$$
(16)$$EC\left(t\right)=E\left(t\right)-E(t-1)$$
(17)$$K_{i}=K_{p}\times \frac{T_{s}}{T_{i}}$$
(18)$$K_{d}=K_{p}\times \frac{T_{d}}{T_{s}}$$

The position-based PID control algorithm used here employs *U*(*t*), which represents the output value of the PID controller at time *t*, *E*(*t*) to represent the deviation at time *t*, *K_p_* as the proportional gain, *K_i_* as the integral gain, *K_d_* as the derivative gain, *T_i_* as the integral time constant, *T_p_* as the derivative time constant, *T_s_* as the sampling time, and *EC*(*t*) as the change in error between the current and previous sampling instances. While a PID controller is a linear controller, it may not effectively address nonlinear system issues. The fuzzy control, a nonlinear controller, can solve nonlinear problems and speed up temperature control while reducing overshoots by adaptively adjusting the PID controller parameters. The temperature control scheme framework is depicted in Figure 4, where two fuzzy controllers, respectively, conduct a nonlinear system adjustment and adaptive modulation. The output *Y*(*t*) at time *t* and target value *R*(*t*) serves as the input for the fuzzy controller, which outputs the PID controller adjustment parameter Δ*K_p_* after undergoing fuzzification, fuzzy inference, and defuzzification. The other fuzzy controller inputs the error *E*(*t*) and its change *EC*(*t*) at time *t*, and outputs Δ*K_i_* and Δ*K_d_* after undergoing fuzzification, fuzzy inference, and defuzzification. After fuzzy logic control, the PID control parameters are expressed as:(19)$$\left\{\begin{array}{cc} K_{p}(t)= & q_{p}\times \Delta K_{p}\\ K_{i}(t)= & K_{i0}+q_{i}\times \Delta K_{i}\times R(t)\\ K_{d}(t)= & K_{d0}+q_{d}\times \Delta K_{d}\times R(t) \end{array}\right.$$
where *q_p_*, *q_i_*, and *q_d_* are the scale factors, and *R*(*t*) is the target value.

### 3.3. Nonlinear Adjustment

This text utilizes a Mamdani-type fuzzy controller. The fuzzy controller transforms the input into computationally processed mathematical expressions through fuzzy theory by utilizing rich human experience. Firstly, the input and output values are mapped to fuzzy subsets through the use of a membership function, such as a triangular, Gaussian, or trapezoidal shape. In this paper, a triangular membership function is used. “Fuzzy subset” is a term used to describe the process of converting clear values of inputs or outputs into fuzzy linguistic values through the use of human language, for example, the letters “B”, “M” and “S” which represent “big”, “medium” and “small”, respectively. The second step is fuzzy logic inference. In this step, fuzzy control rules are established and fuzzy logic inference is performed. For example, “if *R*(*t*) is ‘B’ and *Y*(*t*) is ‘M’, then ∆*K_p_* is ‘S’” where ‘B’, ‘M’, and ‘S’ are used as fuzzy linguistic values. The rules used in this article are more complex. Finally, defuzzification takes place. The output fuzzy quantity is converted into a clear value using a defuzzification method. Methods such as the center of area method, the bisection of area method, and the maximum membership method can be used for defuzzification. This paper uses the center of area method.

Due to the wide temperature range of the controlled object, there is a significant nonlinearity in the system. By conducting experiments, we obtained the input required for the system to reach a steady state at different temperatures, as shown in Figure 5. The analysis shows that the input required for the system to achieve the same temperature change in the low-temperature range is significantly greater than in the high-temperature range, indicating the system’s nonlinearity. During the temperature rise process, both the output value *Y*(*t*) and the target value *R*(*t*) are changing at different time points. By using *Y*(*t*) and *R*(*t*) as inputs to the fuzzy controller and summarizing the fuzzy rules from the experiments, we adjust the proportional coefficient *K_p_* output by the PID controller to accelerate the response speed of the controller in the low-temperature range.

The experiment found that the linearity of the system in the high-temperature range remains basically unchanged. Therefore, in this paper, *Y*(*t*) and *R*(*T*) were taken as input linguistic variables and transformed into ten linguistic values, *T*0, *T*100, *T*150, *T*200, *T*250, *T*300, *T*350, *T*400, *T*600, and *T*800, with a variable range of (0, 800). The triangular membership functions were used and the membership functions are shown in Figure 6a. Fifty-five output linguistic variables of ∆*K_p_* were set and the fuzzy control rule table is shown in Table 2. Based on the set fuzzy control rules, the output surface of ∆*K_p_* was obtained as shown in Figure 7a. In the practical control, ∆*K_p_* is multiplied by a scaling factor to prevent output overshooting and oscillation caused by excessively large proportional coefficients.

### 3.4. Improve Overshoot

The solution to the nonlinear problem of the system has greatly improved the response speed but also brought greater overshoot to the system output. In this paper, The fuzzy control system utilizes *E*(*t*) and *EC*(*t*) as its input variables, and ∆*K_i_* and ∆*K_d_* are used as the outputs of fuzzy control. Linguistic variables are assigned to the inputs, with NB representing negative large, NM representing negative middle, NS representing negative small, ZO representing zero, PS representing positive small, PM representing positive middle, and PB representing positive large (seven linguistic values). The range of these variables was (−6, 6). The triangular affiliation function is used, as shown in Figure 6b. After multiple rounds of experimentation and adjustments, a fuzzy rule table was created and is depicted in Table 3 and Table 4. *E*(*t*) is the difference between the target value and the input value, and *EC*(*t*) is the change in the last adjustment value. For programmed temperature rises, when *E*(*t*) is at zero, it means that the target value and the output value are consistent. If *E*(*t*) is greater than zero, it means that the output value does not reach the target value. If it is less than zero, it means that the output value is greater than the target value and the system has overshoot. If *EC*(*t*) is zero, the temperature remains unchanged. If *EC*(*t*) is greater than zero, the temperature decreases, and if *EC*(*t*) is less than zero, the temperature increases. The fuzzy rule design method is summarized as follows: When the temperature overshoots, but the temperature continues to rise, *K_i_* decreases and *K_d_* increases to suppress the overshoot. When the output value closely matches the target value and the temperature remains relatively stable, *K_i_* and *K_d_* increase, and the time to reach the steady state is accelerated. When the output value is less than the target value, appropriately increase *K_i_* and *K_d_* to speed up the response speed. According to the set fuzzy control rules, the output surfaces of ∆*K_i_* and ∆*K_d_* are obtained as shown in Figure 7b,c.

**Figure 6 micromachines-14-00929-f006:**
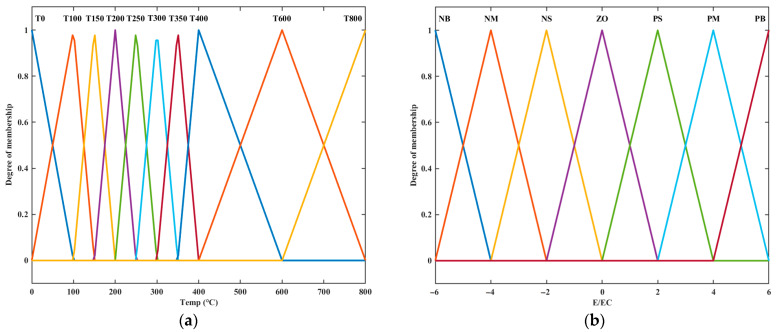
Membership function: (**a**) Membership function of *K_p_*.; (**b**) Membership function of *K_i_* and *K_d_*.

**Figure 7 micromachines-14-00929-f007:**
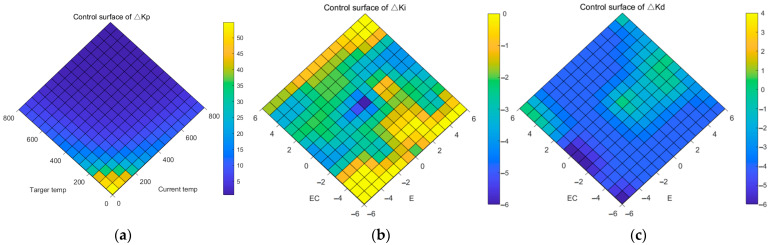
Output surfaces. (**a**) Proportion coefficient; (**b**) Integral coefficient; (**c**) Differential coefficient.

**Table 2 micromachines-14-00929-t002:** Fuzzy control rule table of ∆*K_p_*.

*Y*(*t*)	*R*(*t*)									
	*T*0	*T*100	*T*150	*T*200	*T*250	*T*300	*T*350	*T*400	*T*600	*T*800
*T*0	J1	A1	B1	C1	D1	E1	F1	G1	H1	I1
*T*100	A1	J2	A2	B2	C2	D2	E2	F2	G2	H1
*T*150	B1	A2	J3	A3	B3	C3	D3	E3	F3	G3
*T*200	C1	B2	A3	J4	A4	B4	C4	D4	E4	F4
*T*250	D1	C2	B3	A4	J5	A5	B5	C5	D5	E5
*T*300	E1	D2	C3	B4	A5	J6	A6	B6	C6	D6
*T*350	F1	E2	D3	C4	B5	A6	J7	A7	B7	C7
*T*400	G1	F2	E3	D4	C5	B6	A7	J8	A8	B8
*T*600	H1	G2	F3	E4	D5	C6	B7	A8	J9	A9
*T*800	I1	H2	G3	F4	E5	D6	C7	B8	A9	J10

**Table 3 micromachines-14-00929-t003:** Fuzzy control rule table of ∆*K_i_*.

*E*(*t*)	*EC*(*t*)						
	NB	NM	NS	ZO	PS	PM	PB
NB	ZO	ZO	NM	NM	NS	NM	ZO
NM	ZO	ZO	NM	NS	NS	NM	ZO
NS	ZO	NM	NS	NM	NS	NS	ZO
ZO	NS	ZO	NS	NB	NS	NS	ZO
PS	ZO	ZO	NS	NM	NM	NS	ZO
PM	ZO	NM	NS	NM	NM	ZO	ZO
PB	ZO	NM	NM	NM	NM	ZO	ZO

**Table 4 micromachines-14-00929-t004:** Fuzzy control rule table of ∆*K_d_*.

*E*(*t*)	*EC*(*t*)						
	NB	NM	NS	ZO	PS	PM	PB
NB	NB	NM	NB	NB	NS	ZO	ZO
NM	NM	NM	NM	NM	NM	NM	NM
NS	NM	NM	NM	NM	NM	NM	NM
ZO	NM	NM	NS	ZO	NM	NM	NM
PS	NM	NM	NS	NS	NM	NM	NM
PM	NM	NM	ZO	ZO	NS	NM	NM
PB	NM	NS	NM	ZO	NS	NS	PM

### 3.5. Hardware System Design

Revised for logic and translated to English: In this article, the MCU used is the STM32F105RC. The STM32F105RC is a part of STMicroelectronics’ STM32F1 series of microcontrollers, which are based on the ARM Cortex-M3 processor core. Experimental measurements found that in the high-temperature region (above approximately 800 °C), the temperature changes by 1 °C for every 5 mA change in the current, so a sufficiently precise DAC and ADC are needed for temperature control. For the temperature control section, the 12-bit DAC is utilized to generate analog signals from digital data. The voltage acquisition uses the programmable ADC chip ADS1115 and, during programming, a range of ±4.096 V and an LSB SIZE of 125 μV is used to ensure that the steady-state temperature error in the high-temperature region is controlled within 1 °C, preventing frequent temperature oscillations.

The hardware circuit system includes temperature measurement and temperature control circuits. Due to the dual functions of the temperature measurement and the heating required by the single molybdenum resistor in the design, this article controls the temperature by adjusting the current at both ends of the molybdenum resistor and then measures the voltage across the platinum resistor to calculate the resistance value. The circuit block diagram is shown in Figure 8 below.

The schematic of the constant current source control circuit is shown in Figure 9. The transistor operates in the amplification region, with the MCU controlling the base current, which in turn controls the current through the molybdenum resistor. The current across the molybdenum resistor can be calculated using the potentials on the base and collector, as well as the resistances of *R3* and *R4*. Revised for logic and translated to English: The voltage acquisition circuit employs a design with two voltage followers, followed by a subtractor, which results in more accurate voltage measurements. The schematic is shown in Figure 10.

## 4. Results and Discussions

The temperature control experiments were conducted using traditional PID control, nonlinear adjusted fuzzy PID control, and dual fuzzy PID control to verify the superiority of the temperature control method proposed in this paper. The performance of the control methods was compared by analyzing the simulation and actual system operation data under two conditions: directly setting the temperature to 500 °C and raising the temperature to 500 °C at a rate of 100 °C/s. The experimental data curves and the simulation data curves obtained are shown in Figure 11, Figure 12, Figure 13 and Figure 14.

In the experiment, where the temperature was directly set to 500 °C, the appropriate PID parameters were obtained through experimentation, with *K_p_*_0_ = 1, *K_i_*_0_ = 0.24, and *K_d_*_0_ = 0.05. The simulation was set to run for 2 s with a sampling period of 0.1 s. The data curve obtained from the traditional PID experiment is represented by the black line in the graph. Then, the nonlinear adjustment method mentioned earlier was used to modify the proportional coefficient *K_p_* in the PID parameters using the fuzzy control to increase the response speed. The resulting curve is represented by the red line in the graph. Finally, dual fuzzy control was applied to modify the integral coefficient *K_i_* and differential coefficient *K_d_* in the PID parameters to reduce overshoot. The resulting curve is represented by the blue line in the graph. In the experiment, where the heating rate was given, the duration of the simulation was set to 10 s and the other conditions remained unchanged. The colors of the data curves also remained the same.

This article defines the arrival time *t_arrival_* as the time of the first sample point that is greater than or equal to the target value, and by computing the ratio of the highest overshoot value to the target value, *E_over_* can be ascertained. Comparing the traditional PID control method with the fuzzy PID control method, it can be observed, from Figure 11 and Figure 12, that the control speed of the traditional PID controller is significantly slower than that of the fuzzy PID controller, with a delay of 0.5 s in arrival time, but both simulation data and experimental data show that the overshoot of the fuzzy PID controller is greater than that of the traditional PID controller. The response speed of the dual fuzzy PID controller is similar to that of the fuzzy PID controller, but the overshoot is lower than that of the traditional PID controller. The specific experimental indicators are shown in Table 5.

From Figure 13, it can be clearly observed that in the high-speed temperature control experiment of 100 °C/s, the traditional PID control method is highly non-linear, especially in the low-temperature range, while the fuzzy PID control method and the dual fuzzy PID control method have high linearity. Figure 14 represents the deviation between the target temperature and the current temperature. In the simulation results, the maximum deviation of the traditional PID method can reach 89.3 °C, while the maximum deviation of the fuzzy PID method is 3.2 °C, and the maximum deviation of the dual-mode fuzzy PID method is 3.1 °C. The traditional PID method showed the highest deviation of up to 104.9 °C in practical tests, while the fuzzy PID and dual-mode fuzzy PID methods exhibited deviations of 6.5 °C and 5.7 °C, respectively. The experimental and simulation results are basically consistent and the dual fuzzy control method meets the requirements of the high-speed temperature control.

Finally, a series of experiments were conducted to directly heat from an initial temperature of 50 °C to the target temperature, which was increased in steps of 100 °C from 100 °C to 800 °C, as shown in Table 6. Comparing the results in the table, it can be observed that the traditional PID control method exhibits longer heating times at lower temperatures. In contrast, the fuzzy PID control method and the dual fuzzy PID control method require almost the same time to reach the target temperature. Additionally, the dual fuzzy PID control method outperforms the other methods in terms of overshoot. 

## 5. Conclusions

Traditional TGA faces challenges in achieving high-speed temperature control (About 6000 °C/min). The fastest heating rate for traditional thermogravimetric analysis is 200 °C/min. This paper proposes a dual fuzzy PID temperature control method using a MEMS TGA with on-chip heating, which combines the advantages of the traditional PID control algorithm and the fuzzy PID control algorithm to address the nonlinearity issue in large-span temperature control and effectively suppress overshoot. Simulation and experimental results show that when the temperature control rate reaches 100 °C/s, the maximum error of the dual fuzzy PID control is only 5.7 °C, while the maximum error of traditional PID control reaches 104.9 °C. Furthermore, the overshoot of the dual fuzzy PID control is also reduced compared to the traditional PID control. The temperature control method designed in this paper shows significant advantages over the traditional PID control algorithm and can achieve a temperature control rate that traditional PID control cannot achieve.

## Figures and Tables

**Figure 1 micromachines-14-00929-f001:**
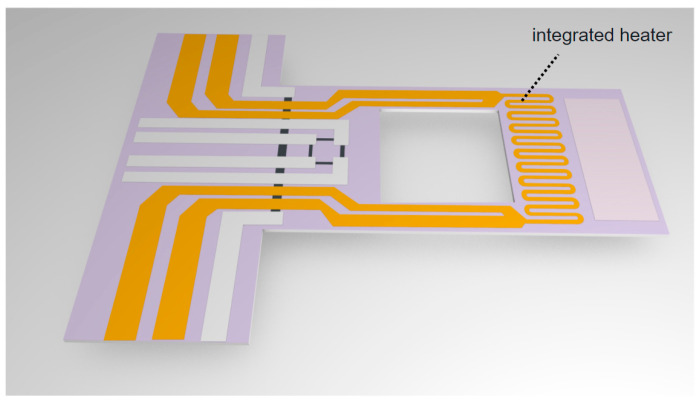
Integrated microheater cantilever.

**Figure 2 micromachines-14-00929-f002:**
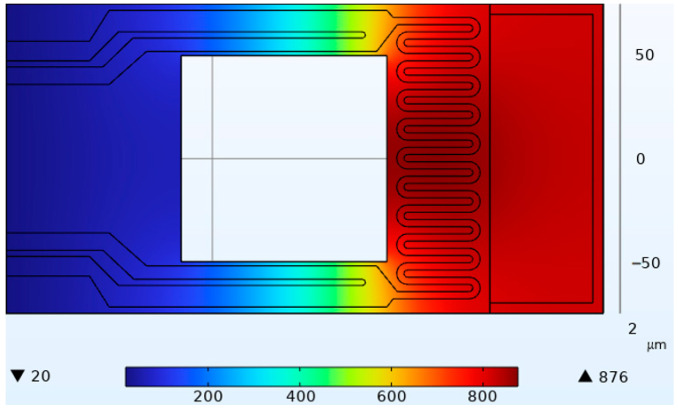
Thermal simulation diagram.

**Figure 3 micromachines-14-00929-f003:**
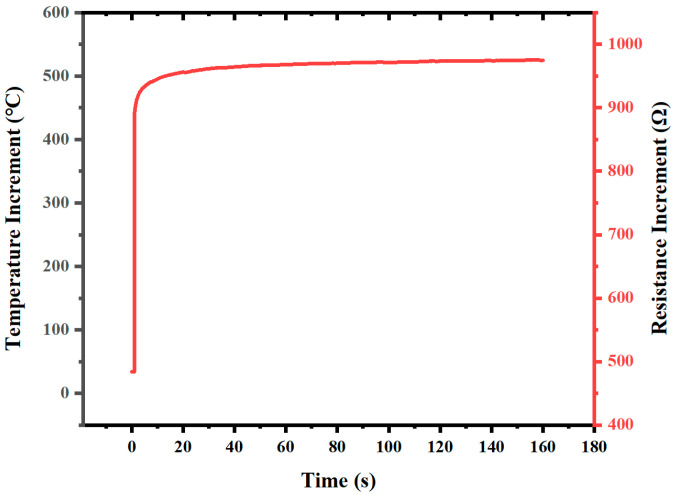
Step response curves.

**Figure 4 micromachines-14-00929-f004:**
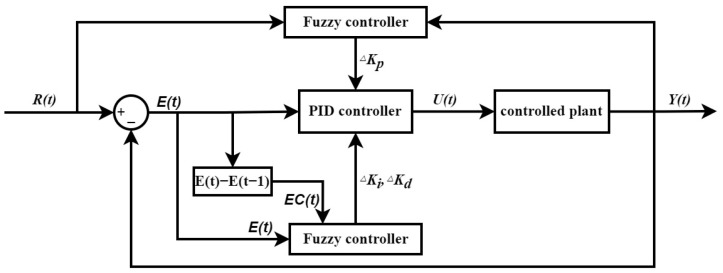
Temperature control system block diagram.

**Figure 5 micromachines-14-00929-f005:**
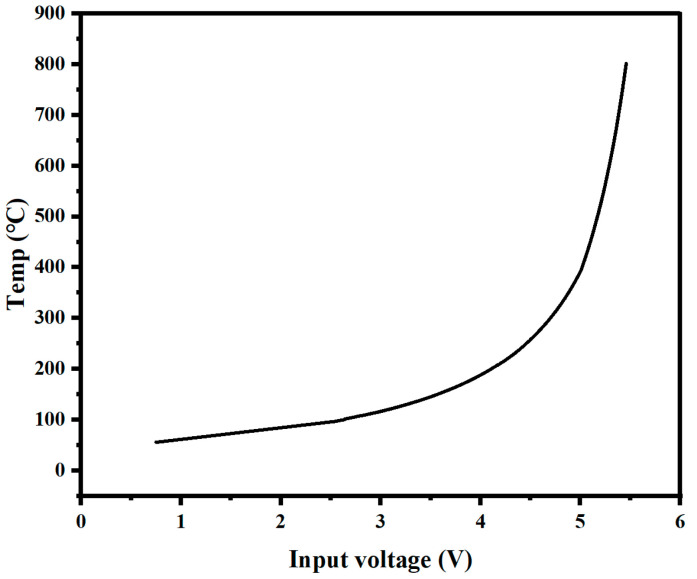
Temperature corresponding to voltage.

**Figure 8 micromachines-14-00929-f008:**
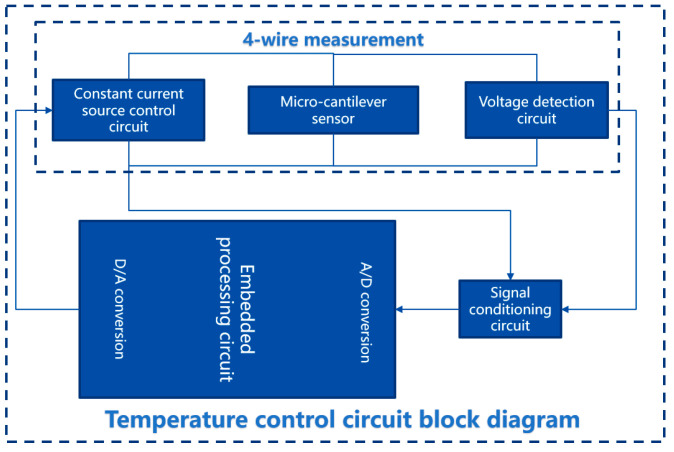
Temperature control circuit block diagram.

**Figure 9 micromachines-14-00929-f009:**
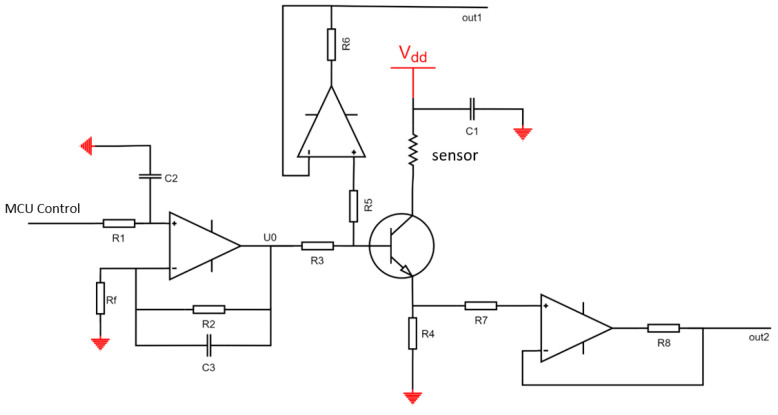
Constant current source control circuit.

**Figure 10 micromachines-14-00929-f010:**
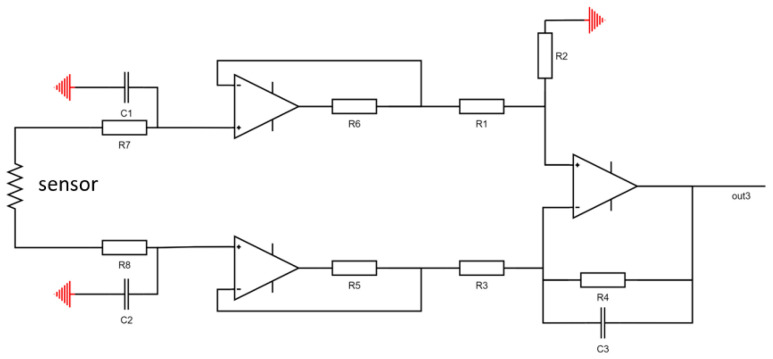
Temperature control circuit block diagram.

**Figure 11 micromachines-14-00929-f011:**
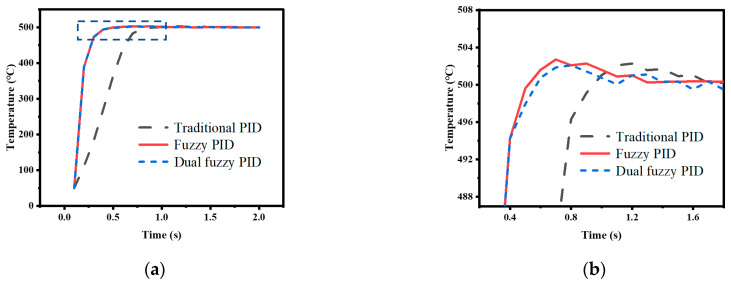
Direct heating to 500 °C test results: (**a**) Simulation data results; (**b**) Bule dashed box part of (**a**) enlarged.

**Figure 12 micromachines-14-00929-f012:**
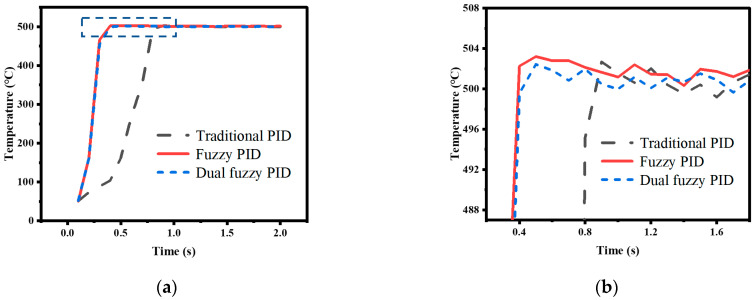
Direct heating to 500 °C test results: (**a**) Actual experimental results; (**b**) Bule dashed box part of (**a**) enlarged.

**Figure 13 micromachines-14-00929-f013:**
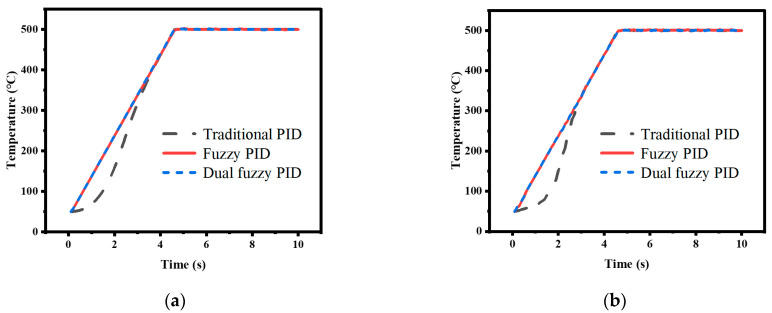
Test results at 100 °C/s to 500 °C: (**a**) Simulation Data Results; (**b**) Actual experimental results.

**Figure 14 micromachines-14-00929-f014:**
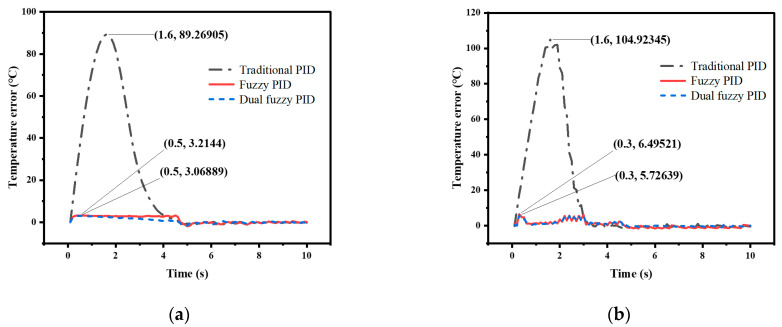
Deviation between the target temperature and the current temperature: (**a**) Simulation Data Results; (**b**) Actual experimental results.

**Table 1 micromachines-14-00929-t001:** Comparison of temperature control rates.

Control Method	Genetic Algorithm-Based Fuzzy PID	Particle Swarm Optimization (PSO)-Based Fuzzy PID	Fuzzy PID Improved Algorithm	Dual Fuzzy PID
Temperature control rate (°C/min)	28	20	30	6000

**Table 5 micromachines-14-00929-t005:** Comparison of three control methods in simulation and experimentation.

Control Method	Simulation		Experiment	
	*t_arrival_* (s)	*E_over_* (‰)	*t_arrival_* (s)	*E_over_* (‰)
Traditional PID	1.0	4.53	0.9	5.35
Fuzzy pid	0.6	5.41	0.4	6.40
Double-fuzzy PID	0.6	4.17	0.5	4.86

**Table 6 micromachines-14-00929-t006:** Comparison of Results of Three Control Methods with Different Target Temperatures.

Temperature Rise at 50 °C	*t_arrival_* (s)			*E_over_* (‰)			Power Consumption (mV)		
	Traditional PID	Fuzzy PID	Double-Fuzzy PID	Traditional PID	Fuzzy PID	Double-Fuzzy PID	Traditional PID	Fuzzy PID	Double-Fuzzy PID
100 °C	3.5	0.6	0.6	2.29	2.67	2.24	2.34	2.62	2.64
200 °C	3.2	0.4	0.5	3.55	5.50	3.15	8.19	7.86	8.70
300 °C	1.9	0.5	0.5	3.92	4.66	3.58	11.66	13.57	13.88
400 °C	1.2	0.4	0.4	3.31	5.03	2.88	11.78	15.77	14.96
500 °C	0.9	0.4	0.5	5.35	6.40	4.86	12.80	17.39	20.14
600 °C	0.7	0.5	0.5	5.10	6.40	4.67	13.01	22.40	21.08
700 °C	0.7	0.4	0.5	6.19	7.84	5.44	13.09	23.91	22.32
800 °C	0.8	0.5	0.5	8.17	8.85	6.06	13.70	25.96	26.69

## Data Availability

The data presented in this study are available on request from the corresponding author.

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
