# Peer review of "High-Speed Temperature Control Method for MEMS Thermal Gravimetric Analyzer Based on Dual Fuzzy PID Control"

_micromachines, 2023, doi:10.3390/mi14050929_

Round 1
Reviewer 1 Report
n this manuscript, X. Z. et al. present an improved method for temperature control of MEMS heaters, leading to a significant enhancement in temperature control rates. The control algorithm has been tested through both simulation and experimentation, yielding interesting results. While the paper is well-organized, I have identified several minor revisions and editing suggestions that could further improve the manuscript:
1. The keywords "temperature control" should have a lowercase first letter, and "fuzzy control" has a font error in the first letter。
2. In line 54, replace "They can be widely used" with "MEMS TGA can be widely used" for clarity.
3. In line 91-92,"reserves a portion of space" should be changed to "reserves a section of space".
4. In the last sentence, change "Molybdenum's resistance" to "The resistance of molybdenum" for clarity
5. In line 104, change "Molybdenum's resistance" to "The resistance of molybdenum" for clarity: "The resistance of molybdenum changes linearly with temperature, which facilitates subsequent temperature control and processing."
6. In line 113,In the first sentence, change "the data from the open-loop step response experiment was imported" to "the data from the open-loop step response experiment were imported."
7. In line 115, add a comma after "PID control" for readability: "PID control, and two fuzzy controllers were used to adjust the PID control parameters in real-time."
8. In line 138, consider replacing "through proportion, integral, and differential components" with "by combining proportional, integral, and differential components.
9. In Table 5, there is a capitalization error in the letters. "Fuzzy pid" should be changed to "Fuzzy PID".
Author Response
Dear reviewer, thank you for reviewing our manuscript and providing valuable feedback. We appreciate your suggestions, which are of great significance to our research. We have provided the following point-to-point answers based on the questions you have raised.
Q1: The keywords "temperature control" should have a lowercase first letter, and "fuzzy control" has a font error in the first letter.
A1: In line 23 We have changed “Temperature control” to “temperature control” in the manuscript. "fuzzy control" The first letter font has been modified.
Q2: In line 54, replace "They can be widely used" with "MEMS TGA can be widely used" for clarity.
A2: In line 54, "They can be widely used" has been modified to "MEMS TGA can be widely used".
Q3: In line 91-92,"reserves a portion of space" should be changed to "reserves a section of space".
A3: In line 96-97, "reserves a portion of space" has been modified to "reserves a section of space".
Q4: In the last sentence, change "Molybdenum's resistance" to "The resistance of molybdenum" for clarity.
A4: In line 108,"Molybdenum's resistance" has been modified to "The resistance of molybdenum".
Q5: In line 104, change "Molybdenum's resistance" to "The resistance of molybdenum" for clarity: "The resistance of molybdenum changes linearly with temperature, which facilitates subsequent temperature control and processing."
A5: Thank you for the review suggestions, In line 109, I have made the necessary revisions.
Q6: In line 113,In the first sentence, change "the data from the open-loop step response experiment was imported" to "the data from the open-loop step response experiment were imported. "
A6: Thank you to the reviewer for their suggestions, In line 123, I have made the corresponding text replacements.
Q7: In line 115, add a comma after "PID control" for readability: "PID control, and two fuzzy controllers were used to adjust the PID control parameters in real-time."
A7: Thank you to the reviewer for the suggestion, In line 125, I have now added the missing punctuation.
Q8: In line 138, consider replacing "through proportion, integral, and differential components" with "by combining proportional, integral, and differential components.
A8: Thank you to the reviewer for your suggestions, In line 160, I have made modifications to the relevant text.
Q9: In Table 5, there is a capitalization error in the letters. "Fuzzy pid" should be changed to "Fuzzy PID".
A9: Thank you for the suggestions from the reviewer, In line 346 I have corrected the errors in the text.
Reviewer 2 Report
The manuscript entitled 'High-speed temperature control method for MEMS thermal gravimetric analyzer based on dual fuzzy PID control' is unable to explain the novelty of the work and requires to improve it a lot. The recommendations and suggestions are embedded in the PDF manuscript attached herewith. It is not recommended to publish to the journal in the present form and maybe reconsider after modifications.

Author Response
Dear reviewer, thank you for reviewing our manuscript and providing valuable feedback. We appreciate your suggestions, which are of great significance to our research. We have provided the following point-to-point answers based on the questions you have raised.
Q1: Is it application of TGA?
A1: Yes, it’s application of TGA.
Q2: Full stop is missing, and what do you mean by etc.
A2: Thank you to the reviewer for your suggestions, I have made modifications to the relevant text. Located in lines 29.
Q3: Unit of temperature is missing.
There is no need of showing the step input signals and hence both the curves can be merged in a single plots with two y axis.
A3: Thank you for the reviewer's suggestion. I have made changes to the relevant images at line 155.
Q4: There is no information for Y axis and what is the difference between above figure 2.
A4: Thank you for the reviewer's suggestion. To avoid any misunderstandings, I have removed the image and added an explanation about the goodness of fit at line 139 in the manuscript.
Q5: What do you mean buy fitting degree?
A5: Thank you for the reviewer's suggestion. I have added an explanation about the goodness of fit from line 139 to 154 in the manuscript.
Q6: It is not looking like a steady state, seems to be increasing after 800℃.
A6: Thank you for the reviewer's comments. I have made the corresponding modifications to the image at line 213 to make it easier to understand. The figure shows the curves of the input voltage and output temperature of the system, reflecting the system's nonlinearity, so it is an upward curve.
Q7: What is the Y axis?
A7: Thank you for the reviewer's comments. I have modified the image at line 213 and added the y-axis for clarity.
Q8: What is the difference between fuzzy PID and dual fuzzy PID control? it look like same.
After 1 sec, all look like identical so, how do you making improvements? What is the impact of 1 sec transient state?
A8: Thank you for the reviewer's suggestion. I have modified the zoomed-in portion of the figures at line 317 and 319 to make the differences between the Fuzzy PID and Dual Fuzzy PID control more visible, especially when reaching steady-state. The new modified figures show that the Dual Fuzzy PID control has a smaller overshoot than the Fuzzy PID control, indicating that it can effectively suppress overshoot. The table also shows that the Dual Fuzzy PID control has the smallest overshoot among all the control methods.
Q9: Can you quantify the high speed temperature?".
A9: Thank you for the reviewer's suggestion. I have made corresponding supplementary explanations at line 348 in the manuscript.
Reviewer 3 Report
Please look at the attached pdf file.

Author Response
Dear reviewer, thank you for reviewing our manuscript and providing valuable feedback. We greatly appreciate your suggestions, as they are of significant importance to our research.
Q1: It is suggested to compare your method with previous works [28-30], and clarify the proposed method advantages in form of a table to justify your novelty.
A1: Thank you for the reviewer's suggestions. I have made some revisions in line 85 of the article, and added a table in line 93.
Q2: 1. The four-wire or Kelvin measurement is a well-known method for accurate resistance measurement. There is no need to explain it here, you can just simply cite a reference. 2. Please explain the fabrication process, the dimensions of the heater (line width, spacing, thickness).
A2: Thank you for the reviewer's suggestions. I have made some revisions to the text, citing references in line 106, and specifying the dimensions of the device and the linewidth, thickness, and spacing of the micro-heater starting from line 100.
Q3: It should be subscript: Rc
A3: Thank you for the reviewer's comments. Corresponding modifications have been made to correct the text error at line 110.
Q4:In equation (1), R0 is resistance at 0°C.But no measurement result is provided for this temperature.Please describe how did you obtained the transfer function without reporting the resistance value at 0°C.
A4: Thank you for the reviewer's comments. Corresponding supplementary explanations for R0 and k have been added to the text from line 111 to 114.
Q5: Is this transfer function the ration of output temperature to input voltage in Laplace domain?
If yes, please describe the effect of input voltage frequency (please plot the frequency response of your transfer function. Is it similar to band stop filter. The transfer function is a little bit confusing, please clarify this issue).
A5: Thank you for the reviewer's comments. Indeed, there were some inaccuracies in the transfer function, which I have now corrected. Since the input voltage in the circuit does not directly act on the integrated heater, it controls the constant current source, which in turn heats the integrated heater. The temperature change of the integrated heater itself causes its resistance to change, making the analysis of the transient response process particularly complex. Therefore, I obtained the measurement results directly through experiments and used the built-in System Identification Toolbox in MATLAB for fitting. The fitting results show that a second-order transfer function with two zeros provides the best fit, so this transfer function is chosen. The magnitude response and phase response of the transfer function are shown in the following figure.
Q6: Please describe why it is sufficient in a quantitative way. For example, how it will affect your proposed model if the fitting accuracy decreases to 80% or increases to 95%. The current explanation is too rough.
A6: Thank you for the reviewer's comments. I have made the necessary modifications to the text and further clarified and explained this section from line 139 to 154.
Q7: Please clarify the unit of temperature.The unit of resistance should be ohm, or its symbol.
A7: Thank you for the reviewer's comments. I have made the corresponding modifications to the image at line 155.
Q8: Please include the units for this figure.
A8: I just read the reviewer's comments. To avoid confusion, I have removed the image and provided corresponding explanations for the goodness of fit from line 139 to 154.
Q9: In equation, Y(t) is output But in equation(4), U(t) is output Please be consistent
A9: Thank you for the reviewer's comments. As there were some unclear expressions in the manuscript leading to misunderstandings, corresponding supplementary explanations have been added at line 163 and 168. As shown in Figure 5 at line 185, U(t) is the output of the PID controller, and Y(t) is the output of the controlled object.
Q10: Unit?
A10: Thank you for the reviewer's comments. I have made the corresponding modifications to the image at line 213.
Q11:
- Please include the schematic of the utilized circuit for your measurement.
- Please include the employed ADC (analog to digital converter) and MCU (micro controller unit) for your measurement.
- Please explain how the noise level and accuracy of the ADC affect your model's performance.
A11: Thank you for the reviewer's comments. I have added Section 3.5 at line 255, which includes the circuit diagram, the models and accuracies of the MCU and ADC used, and an explanation of the impact of accuracy on temperature control.
Q12: Please include the units for all figures.
A12:.Thank you for the reviewer's suggestions. I have made modifications to the images accordingly.
Q13: It is suggested to compare the power consumption of different controller systems.
A13: Thank you for the reviewer's comments. As the sensor controlled by this system has a small volume and heat capacity, the theoretical current of the sensor when heated to the maximum temperature of 1000℃ is approximately 7mA, with a resistance of about 1200Ω, resulting in a power consumption of approximately 0.0588W. This power consumption is much smaller than the device's own power consumption, and the difference in power consumption generated by using different control algorithms is almost negligible. Therefore, it is not suitable to use power consumption as a comparison parameter. Finally, I sincerely hope for your understanding.
Q14: The presented model is based on the resistance variation of the heater. But in real applications the temperature profile at the total area of heater is important. It is suggested to consider this issue in your model.
A14: Thank you for the reviewer's comments. I have added a thermal simulation model of temperature distribution in Figure 2 at line 119.
Round 2
Reviewer 3 Report
Respected authors,
I want to express my gratitude for taking the time to address my comments. I appreciate that you have addressed all of my comments except for Q5 and Q13 mentioned in your response letter.
Regarding Q5, I am seeking clarification on the transfer function in equation (2). Specifically, I am curious whether it represents the ratio of the heater temperature to input current. If this is the case, then the transfer function has two flat regions in the frequency domain, which is unusual for a microheater. Typically, microheaters have only one flat region in the low-frequency range, behaving like a low-pass filter. I request that you clarify this matter in your article since the rest of your analysis is dependent on this transfer function.
As for Q13, I respectfully disagree that 58.8 mW is a small power consumption in the MEMS field. On the contrary, it is quite significant. It is possible that your high heating rate is not solely due to your method, but also because of the high power consumption. If similar works have reported power consumption, I suggest comparing yours with them.
Author Response
Please see the word file attatched.
